# Taiwan’s Smart Healthcare Value Chain: AI Innovation from R&D to Industry Deployment

**DOI:** 10.3390/healthcare14010023

**Published:** 2025-12-21

**Authors:** Tzu-Min Lin, Hui-Wen Yang, Ching-Cheng Han, Chih-Sheng Lin

**Affiliations:** 1Digital Transformation Research Institute, Institute for Information Industry, Taipei 100, Taiwan; rebeccalin@iii.org.tw (T.-M.L.); yang123@iii.org.tw (H.-W.Y.); kevinhan@iii.org.tw (C.-C.H.); 2Department of Biological Science and Technology, National Yang Ming Chiao Tung University, Hsinchu 300, Taiwan; 3Center for Intelligent Drug Systems and Smart Bio-Devices (IDS^2^B), National Yang Ming Chiao Tung University, Hsinchu 300, Taiwan

**Keywords:** Taiwan, artificial intelligence (AI), smart healthcare, information and communications technology (ICT), National Health Insurance

## Abstract

Taiwan’s strategic focus in digital healthcare has been officially integrated into national industrial policy and identified as a crucial application area for artificial intelligence (AI) and next-generation communication technologies. As the healthcare sector undergoes rapid digital transformation, digital healthcare technologies have emerged as essential tools for improving medical quality and efficiency. Leveraging the extensive coverage of its National Health Insurance (NHI) system and its strengths in Information and Communications Technology (ICT), Taiwan also benefits from the robust research capacity of universities and hospitals. Government-driven regulatory reforms and infrastructure initiatives are further accelerating the advancement of the NHI MediCloud system and the broader digital healthcare ecosystem. This article provides a comprehensive overview of smart healthcare development, highlighting government policy support and the R&D capabilities of universities, research institutes, and hospitals. It also examines the ICT industry’s participation in the development of smart healthcare ecosystems, such as Foxconn, Quanta, Acer, ASUS, Wistron, Qisda, etc. With strong data assets, technological expertise, and policy backing, Taiwan demonstrates significant potential in both AI innovation and smart healthcare applications, steadily positioning itself as a key player in the global healthcare market.

## 1. Introduction

Taiwan successfully nurtured remarkable talent in the biopharmaceuticals, electronics, and Information and Communications Technology (ICT) industries. According to the Health Care Index 2024 ranking released by the global database website NUMBEO (https://www.numbeo.com/cost-of-living/ (accessed on 12 November 2025), Taiwan ranked first in the world for eight consecutive years [1], with its world-leading comprehensive healthcare and National Health Insurance (NHI) data assets. Additionally, according to the analysis of the Global Clinical Artificial Intelligence (AI) Dashboard, Taiwan was ranked 10th in the world with 1331 published AI healthcare research papers in 2024, showcasing robust R&D capabilities [2]. In terms of practical implementation, Taiwan effectively integrates some of the key success factors. The revised AI medicine regulations in Taiwan are expected to boost the market of smart healthcare, which is projected to experience significant growth in the coming years [3,4]. This growth is driven by the increasing demand for remote healthcare services and advancements in digital health technologies. It is also fueled by the broader acceptance of AI medicine among healthcare providers and patients.

In Taiwan, the NHI system has amassed over 23 million individual medical records across more than 20 years [5,6]. When combined with the MediCloud platform and electronic medical record (EMR)-sharing mechanism, this system gains enhanced interoperability and data accessibility. These features provide a real-time, comprehensive foundation for AI applications and big data analytics in healthcare [7]. Leveraging the National Health Insurance Administration (NHIA) data assets, strong ICT industry support, and close collaboration among universities and institutions, these efforts have enabled the development of advanced smart hospitals and ICT companies, along with the government’s support. Taiwan is advancing AI applications from research to clinical practice.

Digital-twin technology has emerged as a critical component of modern smart healthcare, enabling the creation of dynamic, patient-specific virtual models that mirror physiological states in real time. These models support predictive diagnostics, personalized treatment planning, and proactive monitoring of disease progression, thereby complementing AI and big-data approaches [8,9]. Recent advances include cardiovascular digital twins for risk prediction and surgical planning systems that simulate patient anatomy, and hospital workflow optimization through digital replicas of clinical processes [10,11]. Taiwan’s smart healthcare ecosystem has begun exploring digital-twin applications, particularly in precision medicine and hospital resource management. However, integration remains at an early stage compared to leading international initiatives [12]. Incorporating digital-twin healthcare into the state-of-the-art review ensures a more comprehensive account of essential technologies shaping the future of smart healthcare.

This review was conducted as a structured narrative analysis of Taiwan’s smart healthcare value chain, focusing on AI innovation from research and development to industry deployment. Sources were identified through multiple channels, including academic databases (PubMed, Scopus, Web of Science, and Google) and government portals such as the Taiwan Food and Drug Administration (TFDA) and Ministry of Health and Welfare (MOHW). They also include industry reports and gray literature, including conference proceedings and news archives [13]. The time frame covered was 1995–2025, reflecting both recent developments and forward-looking initiatives. Inclusion criteria were limited to TFDA-approved AI medical devices [14], national-level programs and policies [15], and major ICT company initiatives with documented healthcare applications [16]. Exclusion criteria included pilot projects without regulatory approval, institution-specific programs lacking national relevance, and speculative concepts without published evidence. Data were synthesized to highlight regulatory frameworks, institutional achievements, and industrial contributions, with cross-referencing to international benchmarks, such as Software as Medical Device (SaMD) guidance of the United States (U.S.) Food and Drug Administration (FDA) [17] and the European Union (EU) AI Act [18]. This approach ensures transparency in source selection and provides a comprehensive yet methodologically grounded landscape review.

## 2. Timeline of the Development of Smart Healthcare in Taiwan

The development of Taiwan’s smart healthcare accelerated at the end of the 2010s, driven by government policy, a strong ICT infrastructure, and a universal-hospital healthcare system [19]. Figure 1 and Table 1 demonstrate a concise, date-stamped timeline of how smart healthcare has developed in Taiwan. It covers policy rails, data infrastructure, regulation, and major hospital milestones. Taiwan’s smart healthcare development has progressed through a series of policy milestones that established the foundation for AI integration. The launch of the NHI system in 1995 created a unified data backbone, followed by early EMR integration in the 2000s and the establishment of the EMR Exchange Center in 2011 [20]. Subsequent initiatives included PharmaCloud (2013) and MediCloud (2015), which expanded access to clinical data nationwide. In 2017, precision medicine was identified as a strategic industry, and the AI Taiwan Action Plan (2018) marked the first national push for AI adoption. Regulatory breakthroughs came in 2020 with TFDA’s approval of the first AI-assisted diagnostic system, while telemedicine reforms during COVID-19 in 2022 accelerated virtual care [21,22]. More recent milestones include the release of AI Taiwan Action Plan 2.0 (2023) [23,24], revisions to telehealth regulations (2024), and NHIA’s five-year AI program with Google Health. They also encompass hospital achievements such as Healthcare Information and Management Systems Society (HIMSS) Stage 7 certifications in 2025, alongside MOHW’s drive for Fast Healthcare Interoperability Resources (FHIR) standardization. Together, these developments illustrate a clear chronological trajectory from foundational infrastructure to advanced digital maturity.

## 3. Comparative Synthesis of Major National Programs

Table 2 shows the major programs driving smart healthcare development in Taiwan since 2019. These program themes are nationally led and implemented by the NSTC or the MOHW. Universities, research institutions, and smart healthcare companies across Taiwan are widely invited, individually or collaboratively, to propose unique and significant thematic projects supported by the government (Figure 2). The ultimate goal is to transform Taiwan into an “AI-powered island” and advance the national vision of a “Healthy Taiwan” [40,41].

### 3.1. Major Programs Driving Smart Healthcare Development in Taiwan Since 2019

Taiwan’s smart healthcare ecosystem is propelled by several national programs, each with distinct project duration and funding, distinctive features, and measurable outcomes to date (October 2025) (Table 2) [42,43,44,45]. The Taiwan Precision Medicine Initiative (TPMI; launched in 2019) builds a genomic and EMR database with over 560,000 participants [46,47,48], filling a global gap in Han Chinese representation. The NSTC smart healthcare Program Phase I (2020–2023) emphasized technological exploration, producing 23 cross-hospital projects and eight TFDA-approved AI devices [49,50]. Phase II (2024–2027) shifts toward patient-centric modular systems such as AI-assisted lung cancer diagnosis [51,52]. The Taiwan smart healthcare Industry-Academia Alliance (TSHA; launched 2022) enables cross-institutional validation, with 23 AI teams demonstrating translational impact in areas like acute kidney injury prediction [53,54]. The Next-Generation Digital Health Platform (DHP; 2024–2027) modernizes infrastructure through FHIR standards, rule libraries, and SMART on FHIR applications. It was further supported by three AI centers for responsible implementation, validation, and impact research. Finally, the Healthy Taiwan Deepening Project (2025–2029) invests USD 1.63 billion to strengthen NHI sustainability, workforce training, as well as Environmental, Social, and Governance (ESG)-aligned healthcare. Synthesizing these programs highlights Taiwan’s balanced strategy: building data assets, fostering innovation, enabling clinical integration, and ensuring long-term sustainability.

### 3.2. Integrated Comparison of the National Programs

To strengthen the assessment of Taiwan’s smart healthcare initiatives, it is important to situate achievements within structured evaluation frameworks and international benchmarks. At the clinical level, hospital digital maturity is increasingly measured against the HIMSS validation models. Taiwan’s flagship hospitals are achieving Stage 7 in both EMRAM and AMAM, demonstrating alignment with global standards of interoperability and analytics [55]. Regulatory progress can be evaluated through comparison with international frameworks such as the EU AI Act, which emphasizes risk-based classification and transparency [18]. The U.S. FDA’s SaMD guidance introduces lifecycle management tools like Predetermined Change Control Plans [17,19]. Taiwan’s TFDA approvals of AI-assisted diagnostic systems provide evidence of regulatory maturity (Table 3) [56]. The establishment of Responsible AI centers and FHIR profile initiatives highlights convergence with international best practices [57]. Industrialization outcomes can be benchmarked against ICT export capacity and commercialization programs, showing parallels with EU CE-mark pathways [18] and FDA market authorizations [19]. By applying these evaluation lenses, Taiwan’s achievements are not only documented but also contextualized within global trajectories, underscoring both its unique strengths and opportunities for further alignment. This comparative approach ensures that Taiwan’s smart healthcare development is assessed not only by domestic milestones but also by its contribution to international standards and sustainable health governance.

## 4. The University-Led Achievements Driving Taiwan’s Smart Healthcare Industry

Taiwan’s universities have played a pivotal role in driving the nation’s smart healthcare industry by serving as incubators of cutting-edge research and innovation (Table 4). Institutions such as National Taiwan University (NTU), National Yang Ming Chiao Tung University (NYCU), National Cheng Kung University (NCKU), Taipei Medical University (TMU), China Medical University (CMU), Kaohsiung Medical University (KMU), Chang Gung University (CGU), etc., have led in fields ranging from AI-driven diagnostics to biomedicine and digital therapeutics. Their academic rigor has laid the groundwork for new technologies that improve early disease detection, patient monitoring, and precision medicine. These research breakthroughs not only enrich academic knowledge but also directly fuel practical applications within the healthcare ecosystem. A hallmark of Taiwan’s smart healthcare growth has been the strong collaboration between universities and industry partners [62]. By forming strategic alliances with hospitals, startups, and technology companies, universities have transformed laboratory findings into scalable healthcare solutions. For example, AI imaging tools developed in university labs have been commercialized for use in hospitals nationwide. These tools have significantly improved diagnostic accuracy and reduced medical workloads [63]. Such partnerships also create channels for knowledge transfer, ensuring that Taiwan’s healthcare system benefits from the rapid integration of advanced research outcomes.

## 5. Key Taiwan’s ICT Players in the Smart Healthcare Industry

The strong ICT industry in Taiwan is contributing to smart healthcare solutions across a diverse range of sub-sectors [27,78]. These include biomedical key components, medical devices, mobile healthcare, gene and cell therapies, as well as smart hospital solutions. The government has adopted a strategic framework organized around four core pillars. They comprise data integration, AI/ICT innovation, cross-hospital collaboration, and the industrialization of medical devices. The output value of Taiwan’s smart healthcare industry in 2024 is estimated to be USD 2 billion [79]. Moreover, the total output value of Taiwan’s ICT industry ranks second in the world, after the USA. By leveraging Taiwan’s strengths in clinical medicine and ICT industries, this competitive edge is expected to catalyze Taiwan’s next industrial boom.

Some of the major electronics and ICT hardware companies in Taiwan are leading the way in system integration for healthcare [79,80,81] (Table 5). Quanta Computer has developed the QOCA telemedicine platform [82], playing a critical role in enabling remote healthcare services in smart hospitals. ASUS offers the AICS smart healthcare solution, a smart healthcare solution integrating AI and the Internet of Things (IoT) to support clinical environments [83]. Acer Medical (a branch of Acer) developed the AI diagnostic tools VeriSee DR (for diabetic retinopathy) and VeriSee AMD [84,85], as well as its TeleMed telemedicine app, enabling remote patient consultation and monitoring. BenQ Medical Technology (via Qisda) develops advanced medical hardware critical to smart hospital systems [86]. Leadtek expanded its products into healthcare with oximeters (Alvital brand), wearable sleep monitoring services, and home health products. Wistron Medical Technology (via Wistron) produced the BestShape VS system using a non-contact vital signs sensor to detect the elderly’s respiration, heart rate, and not-in-bed status without touching them [87,88]. It provides 24 h continuous real-time data display and monitoring. Compal Electronics has created the Compal iCare System and BoostFix. Compal’s iCare platform currently supports over 3000 long-term and residential care institutions across Taiwan, handling administrative tasks like uploading data to the NHI Bureau [89,90]. Leading EMS (Electronics Manufacturing Service) firms, Foxconn, Quanta, Acer, ASUS, Acer, Qisda, Inventec, Wistron, etc., are deepening their investments in smart healthcare, recognizing strong synergies between ICT advancements and healthcare innovation [91,92,93,94].

In addition to the ICT companies mentioned above, Industrial Technology Research Institute (ITRI) plays a pivotal role by supporting smart medical innovation and startups through its StarFab accelerator. ITRI provides cross-industry R&D, infrastructure, and incubation resources for ICT-healthcare integration [95,96,97].

## 6. AI-Assisted Products in Taiwan’s Smart Healthcare Industry

Taiwan’s smart healthcare sector is undergoing rapid advancement, powered by firms producing healthcare AI and related products. Their innovations improve diagnostic accuracy, streamline clinical workflows, and enable more personalized patient care [97]. With strong government support, a robust ICT ecosystem, and rich health data from its NHI system, Taiwan has become a fertile ground for AI innovation in medicine [19,98]. The TFDA has been actively approving a range of smart medical devices and systems that integrate advanced technologies such as AI/machine learning (AI/ML) and computer-aided detection and diagnosis [99,100,101]. This section explores key AI-related products currently deployed across hospitals, clinics, and digital health platforms (Table 6).

Taiwan’s AI healthcare products are not futuristic prototypes. They are clinic-ready tools designed to improve throughput, accuracy, and patient experience. The success of these products depends on four factors. There are abundant NHI health data, strong hospital–tech company partnerships, flexible regulation under the TFDA and Personal Data Protection Act (PDPA), and a practical design philosophy where AI supports clinicians rather than replaces them.

Several locally developed AI-powered systems have successfully obtained certification from the TFDA, marking a significant milestone in clinical innovation. One notable example is VeriSee DR, an AI-powered diabetic retinopathy screening tool approved by Taiwan’s TFDA [84]. Another standout is AmCAD-UT, an AI ultrasound analysis for thyroid cancer risk stratification [102,103,104]. The aetherAI developed a device using digital morphology analysis software for bone marrow smear evaluation. It is an AI colonoscopy with clinical performance reported at 96% accuracy [105]. NCKU and Megapro Biomedical co-developed an iMbody that, as a sarcopenia screening test, predicts mortality among hospitalized cancer patients [106]. An AI-assisted lung cancer diagnosis module was developed by V5med Inc. and AstraZeneca Taiwan. Starting with low-dose CT image interpretation, the module supports early lesion detection of solitary pulmonary nodules [107]. The BestShape VS system, developed by Wistron Medical Technology, is a non-contact vital signs sensor to detect the elderly’s respiration, heart rate, and not-in-bed status without touching them [87]. Additionally, Fubon Insurance in Taiwan has developed a smart AI claims review system through which simple claims can be processed automatically to allow express payouts. These success stories reflect Taiwan’s maturity in AI medical research and the government’s proactive support through regulatory guidance. As TFDA continues to refine its review standards for AI/ML-based medical devices or systems, more locally developed systems are expected to enter clinical practice. This offers a compelling model for other nations seeking to balance technology, policy, and patient-centered care.

Based on the information compiled in Table 6, we categorized the technologies by their underlying principles and application domains into a six-category matrix so readers can quickly grasp the current development of smart healthcare technologies and products in Taiwan (Figure 3). The six categories cover complete clinical scenarios, including “Clinical diagnosis”, “Data integration”, “Intelligent assistants”, “Ward care”, “Telemedicine”, and “Administrative management” The message indicates that the development of AI medical devices in Taiwan has progressed from edge AI technology to a comprehensive smart healthcare ecosystem.

## 7. Taiwan’s Smart Healthcare Development Pathway 

Taiwan’s smart healthcare value chain is being established by sequential progression from data infrastructure. This progression extends to population-level health impact within a national smart healthcare ecosystem (Figure 4). Data Foundations is the first established by nationwide platforms such as NHI MediCloud, EMR, and PharmaCloud. R&D and validation encompass the advancement of AI-driven innovation through TPMI and the NSTC Smart Healthcare Program Phase I. Regulation and standards reflect Taiwan’s governance mechanisms that incorporate TFDA approval pathways and MOHW-led FHIR standardization [110]. Commercialization and industrialization highlight the expansion of the smart healthcare sector through ICT industry engagement and SHIrT (Sustainable, High-tech, Innovative, Resilient, and Transformative) investment [111]. Clinical Integration demonstrates the deployment of digital tools across care settings by achieving HIMSS Stage 7 digital maturity and telehealth expansion [37]. Outcomes and societal impact capture the system-level benefits realized through improved patient safety, enhanced equity of access, and strengthened long-term sustainability.

This figure shows Taiwan’s stepwise progression in building a national smart healthcare ecosystem. It begins with data foundations (NHI MediCloud, EMR, PharmaCloud), advances through AI R&D and validation (TPMI, NSTC Phase I), and incorporates regulation and standards (TFDA approvals, MOHW-led FHIR). The pathway continues with commercialization and industrialization via ICT industry engagement, followed by clinical integration through HIMSS Stage 7 maturity and telehealth expansion. Ultimately, these efforts deliver societal impact, improving patient safety, equity of access, and long-term sustainability.

## 8. Taiwan’s Smart Healthcare Ecosystem: Drivers, Infrastructure, and Applications

Taiwan’s prioritization of smart healthcare is strategically driven by pressing demographic and systemic challenges, including one of the fastest rates of population aging in Asia, with citizens aged 65 and above projected to exceed 30% by 2039. It is also shaped by persistent workforce shortages among nurses and emergency staff, alongside a high burden of chronic diseases such as diabetes, cardiovascular disorders, kidney disease, and cancer [112,113,114,115]. To address these pressures, Taiwan’s NHI system provides a uniquely integrated data architecture—combining claims data, the MediCloud clinical hub, and patient-accessible My Health Bank records. This architecture creates a longitudinal dataset covering more than 23 million individuals and enabling real-world evidence generation, federated learning, and cross-hospital interoperability [28,34,38]. Building on this foundation, hospitals have deployed AI-enabled applications with tangible clinical impact. These include predictive analytics for acute kidney injury, TFDA-approved imaging diagnostics for diabetic retinopathy and lung cancer, and expanded telemedicine platforms during the COVID-19 pandemic [32,35,66]. These advances are supported by a coordinated continuum linking government policy, academic research, and ICT industry collaboration. National programs such as the NSTC smart healthcare initiatives fund AI model development and validation, and ICT firms, including Acer, Quanta, and Wistron, drive commercialization and deployment [49,64,78]. Clarifications of MediCloud’s role as a nationwide hub integrating laboratory, imaging, prescription, and discharge records, alongside NHI’s data-sharing mechanisms, are essential. These clarifications further improve readability for international audiences [7,28]. Nonetheless, barriers remain, including challenges in data governance, uneven adoption of interoperability standards such as FHIR, limited regulatory sandbox scope, and the need for harmonized AI validation guidelines [32,66]. Looking outward, Taiwan’s strong ICT sector creates opportunities for cross-border collaborations in AI healthcare. It leverages expertise in semiconductors, cloud infrastructure, and AI hardware accelerators to co-develop imaging diagnostics, predictive analytics, and telemedicine solutions with global partners [27,78]. Finally, a patient-centric approach that emphasizes privacy protection, cybersecurity, and equitable access to smart healthcare technologies remains essential. This approach sustains trust, inclusivity, and international credibility in Taiwan’s digital health transformation [40,116].

## 9. Limitations and Future Directions

Despite notable achievements, Taiwan’s AI medical products face persistent challenges that limit global competitiveness. Many models are trained primarily on NHI datasets, which, while comprehensive domestically, raise reproducibility concerns when applied to diverse international populations [32,50]. Regulatory approvals by TFDA have accelerated local adoption, yet limited alignment with frameworks such as the U.S. FDA SaMD Action Plan and the EU AI Act constrains international recognition [17,18]. Furthermore, commercialization strategies remain underdeveloped, with insufficient global clinical trials and cross-border partnerships to validate scalability. These issues underscore that technological maturity alone is not sufficient; without harmonized regulation, reproducibility across populations, and stronger international market strategies, Taiwan’s AI medical products will continue to struggle in achieving robust global competitiveness.

Several limitations remain that highlight important avenues for future directions to advance smart healthcare in Taiwan. First, data privacy and security concerns persist, particularly in the context of large-scale NHI datasets and cross-institutional sharing. While initiatives such as the Next-Generation Digital Health Platform emphasize federated learning and distributed training [117], further research is still needed. This research should strengthen privacy-preserving technologies, including homomorphic encryption and blockchain-based medical record systems. Second, regulatory and standardization gaps continue to challenge interoperability. Despite ongoing efforts to adopt FHIR and harmonize AI medical device approval processes [38], Taiwan must accelerate the development of unified standards and regulatory pathways. These pathways should balance innovation with patient safety. Third, many AI applications remain in pilot stages, with limited large-scale clinical validation. This raises questions about generalizability beyond Taiwan’s Han Chinese population, underscoring the importance of international collaboration and comparative studies [46,47,48]. Fourth, workforce readiness and uneven adoption across hospitals remain barriers. Future studies should investigate effective training models, change-management strategies, and user-centered design to ensure clinicians can integrate AI seamlessly into daily workflows [118]. Fifth, economic sustainability requires further evaluation of reimbursement models and cost-effectiveness, particularly for smaller hospitals and rural clinics. Programs such as the Smart Healthcare Innovation and Entrepreneurship Investment Program [116] provide financial support, but long-term viability will depend on clear evidence of clinical and economic value. Finally, ethical and social considerations, including algorithmic transparency, accountability, and equitable access, must be addressed to ensure that AI healthcare benefits all socioeconomic groups [33,118]. Addressing these challenges through targeted research will not only strengthen Taiwan’s domestic smart healthcare ecosystem but also enhance its role as a global contributor to responsible and sustainable AI innovation in medicine. In this way, limitations become opportunities for Taiwan to refine its policies, expand its research agenda, and set benchmarks for international best practices.

In addition to Taiwan’s national programs and ICT-driven initiatives, recent advances in computational methods further enrich the smart healthcare value chain. For example, secure data transmission remains critical for medical IoT applications. Novel cryptographic approaches such as the Dynamic Hill Cipher have been proposed to enhance privacy and resilience in healthcare networks [119]. At the biomedical research level, improved algorithms for aligning protein–protein interaction networks, such as NAIGO, demonstrate their effectiveness. These graph-based and ontology-driven methods can accelerate translational insights and precision medicine [120]. Meanwhile, in the domain of clinical text mining, innovative deep learning models have advanced Chinese medical named-entity recognition. These models support more accurate extraction of patient information from electronic health records [121]. Together, these developments highlight the complementary role of secure data infrastructure, bioinformatics, and natural language processing in strengthening Taiwan’s smart healthcare ecosystem. They also position Taiwan within global AI innovation trends.

## 10. Conclusions

Taiwan’s smart healthcare ecosystem has achieved notable progress in recent years. This progress is reflected in hospital digital maturity certifications such as HIMSS EMRAM and AMAM Stage 7. It is also evident in regulatory approvals of AI/ML medical devices by the TFDA and the integration of NHI big-data resources into AI model development. These achievements demonstrate Taiwan’s capacity to combine a strong public health infrastructure with emerging digital technologies. However, several gaps remain. These include governance challenges in aligning TFDA regulations with international frameworks such as the FDA SaMD Action Plan and the EU AI Act. Additional issues include interoperability limitations across hospital systems and commercialization hurdles for ICT-driven healthcare solutions. Reproducibility also remains a critical issue, as many AI models trained on localized datasets face difficulties in generalizing to diverse populations. Addressing these gaps will require stronger evaluation frameworks, harmonized regulatory standards, and enhanced collaboration between government, academia, and industry. Forward-looking strategies should prioritize reproducibility testing, cross-border regulatory recognition, and investment in commercialization pathways. These efforts will ensure that Taiwan’s innovations can scale globally. In addition, leveraging the WHO’s Global Benchmarking Tool can help situate Taiwan’s regulatory maturity within international comparisons and guide capacity building. By combining achievements with critical reflection on limitations, Taiwan can strengthen its position as a leader in smart healthcare while ensuring sustainable and globally relevant outcomes.

## Figures and Tables

**Figure 1 healthcare-14-00023-f001:**
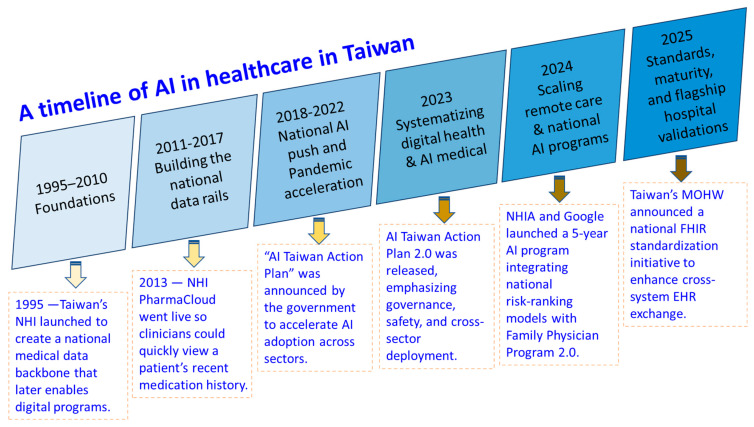
A timeline of AI in healthcare in Taiwan. NHI, National Health Insurance; EMR, Electronic Medical Record; MOHW, Ministry of Health and Welfare; NHIA, National Health Insurance Administration; FHIR, Fast Healthcare Interoperability Resource.

**Figure 2 healthcare-14-00023-f002:**
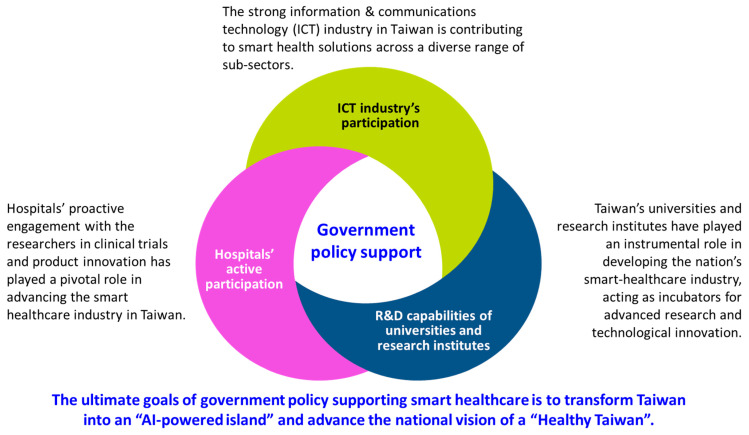
Collaborative research and development among key stakeholders are driving the industrialization of smart healthcare in Taiwan.

**Figure 3 healthcare-14-00023-f003:**
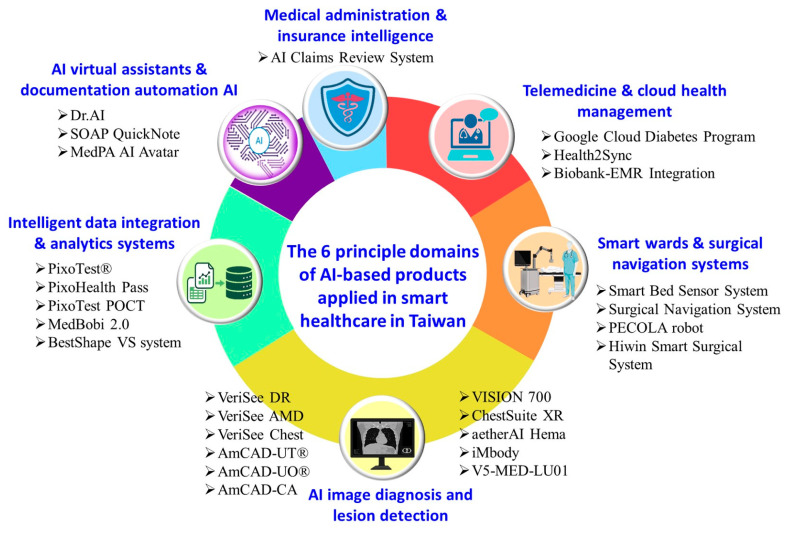
The principle and application domains of AI-based products applied in smart healthcare in Taiwan.

**Figure 4 healthcare-14-00023-f004:**
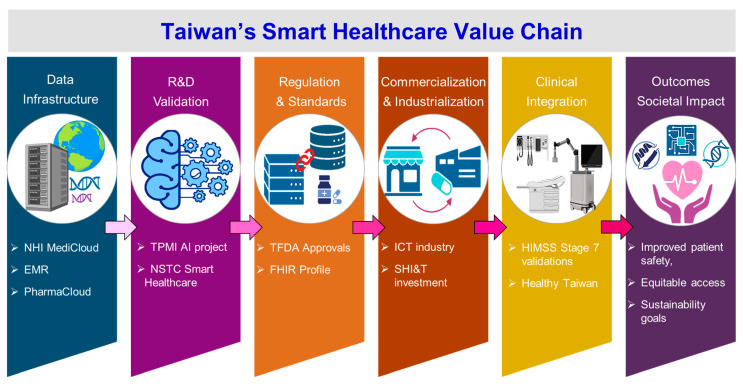
Taiwan’s smart healthcare value chain.

**Table 1 healthcare-14-00023-t001:** Date-stamped timeline of the development of smart healthcare in Taiwan.

Category	Stages (Duration)	Key Implementation Highlights	Reference
Data Infrastructure	Foundations (1995–2010)	1995—Taiwan’s NHI was launched to create a single-payer system and national data backbone that later enabled digital/AI programs.	[25]
		2001—To support future AI model development and big data analytics in healthcare, NHIA initiated efforts to integrate ERC with hospital information systems.	[26]
		2005—In alignment with the WHO’s eHealth framework, Taiwan progressively integrated ICT into healthcare services and commenced strategic planning for digital health policies.	[27]
	Building the national data rails (2011–2017)	2011—MOHW set up the ERC Exchange Center to let hospitals exchange EMRs nationwide (initially using HL7 CDA R2).	[28]
		2013—NHI PharmaCloud went live so clinicians could quickly view a patient’s recent medication history.	[29]
		2015—NHI MediCloud expanded PharmaCloud into a multi-registry clinical data hub (labs, imaging, surgeries, allergies, discharge summaries, etc.).	[7]
		2017—Precision medicine has been featured as one of the key industries in the Biomedical Industry Innovation Program launched, focusing on pharmaceuticals, medical devices, applied biotechnology, and health and well-being.	[30]
Regulation	First national AI push, pandemic acceleration, and early telemedicine easing (2018–2022)	“AI Taiwan Action Plan” was announced by the Executive Yuan of Taiwan to accelerate AI adoption across sectors, including healthcare.	[31]
		2020—TFDA approved the first AI-assisted diagnostic system for diabetic retinopathy interpretation, marking the official entry of AI-based medical devices into clinical application.	[32]
		2022—MOHW relaxed telemedicine interpretation during COVID-19, catalyzing virtual-care pilots.	[19]
	Systematizing digital health and AI medical (2023)	Apr. 2023—AI Taiwan Action Plan 2.0 was released, emphasizing governance, safety, and cross-sector deployment (healthcare included).	[33]
		Nov. 2023—NHIA ↔ HIMSS Memorandum of understanding (MOU) aligned with global standards, cybersecurity, and workforce upskilling for digital health.	[34]
Hospital milestones	Scaling remote care and national AI programs (2024)	2024—Telehealth regulations revised to ease e-prescriptions and allow teleconsultation in more circumstances.	[35]
		Jun. 2024—NHIA ↔ Google Cloud/Google Health 5-year AI program (diabetes first): national risk-ranking models were integrated with Family Physician Program 2.0.	[36]
	Standards, maturity, and flagship hospital validations (2025)	Mar. 2025—The hospital achieved HIMSS Analytics Maturity Adoption Model (AMAM) Stage 7. CMUH previously scored # 1 globally on the HIMSS Digital Health Indicator 2023.	[37]
		Mar. 2025—MOHW signaled a national FHIR standardization drive, aiming for smoother cross-system EHR exchange (FHIR-based Taiwan profiles).	[38]
		Jul. 2025—The hospital validated at the HIMSS Electronic Medical Record Adoption Model (EMRAM) Stage 7, underscoring hospital-level digital maturity momentum.	[39]

**Table 2 healthcare-14-00023-t002:** Major national programs driving smart healthcare development in Taiwan.

Title of Program	Project Duration & Funding	Distinctive Features	Measurable Outcomes to Date
The Taiwan Precision Medicine Initiative (TPMI)	2019–present NSTC; Academia Sinica, in collaboration with 16 major medical centers around Taiwan	As of the end of December 2023, 565,390 participants have been recruited for DNA profiling and EMR access	Large biobank established >500 k genomic datasets linked with EMR Multiple precision medicine studies initiated
Phase I of the NSTC Smart Healthcare Program—“The interdisciplinary development and technology project using AI and clinical database”	2020–2023 NSTC	Cross-sector AI projects, collaboration among medical, academic, and industrial partners to advance multiple cross-hospital AI and smart healthcare projects	>20 AI prototypes validated in hospitals; several TFDA approvals achieved
Phase II of the NSTC Smart Healthcare Program—“Taiwan Smart Healthcare Innovation and Value-added Promotion Program”	2024–2027 NSTC	Focus on “Smart Clinical Decision Support” and “GenAI-powered Smart Hospital Management”	Nine modular systems under development; early pilots in cardiovascular and oncology domains
Taiwan Smart Healthcare Alliance (TSHA)	2022–present NSTC	To strengthen AI applications in healthcare, Taiwan mobilized leading medical centers to form a national AI task force under the smart healthcare academia-industry alliance program	Since 2022, 23 AI teams have been validating solutions Multiple AI models deployed (e.g., brain metastasis diagnosis, AKI risk prediction)
Smart Healthcare Innovation and Entrepreneurship Investment Program	2024–2027 National Development Fund, Executive Yuan of Taiwan USD 330 million	Financial support for enterprises ESG-aligned smart hospital vision	>30 enterprises funded Early commercialization of AI health solutions
Next-Generation Digital Health Platform (DHP)	2024–2027 MOHW	Initial FHIR adoption in pilot hospitals; establishment of AI validation centers FHIR standards, Rule Library, SMART on FHIR apps	Creating three AI-based centers for smart healthcare: the Responsible AI Implementation Center, the AI Validation and Certification Center, and the AI Impact Research Center.
Healthy Taiwan Deepening Project	2025–2029 MOHW USD 1.63 billion	ESG-aligned healthcare modernization	Workforce training programs launched; smart hospital upgrades underway

**Table 3 healthcare-14-00023-t003:** Comparison of Taiwan’s major smart healthcare programs and international benchmarks.

Program/Framework	Goals	Funding/Resources	Current Status	Outcomes/Benchmarks
Taiwan—smart healthcare Promotion Program (MOHW, 2018–2025) [58]	Digital transformation of hospitals; integration of AI in clinical workflows	National MOHW budget allocations; ICT industry partnerships	Ongoing; multiple hospitals achieving HIMSS EMRAM/AMAM Stage 7	Improved hospital digital maturity; pilot AI deployments in imaging and chronic disease management
Taiwan—TFDA AI/Machine Learning (AI/ML) Medical Device Approval Initiative (2020–2025) [14]	Regulatory pathway for AI/ML medical devices	TFDA regulatory resources; collaboration with academic R&D	Approved >30 AI/ML devices (diagnostics, imaging, ophthalmology)	National approvals; limited international recognition; need for FDA/CE alignment
Taiwan—NHI Big Data Applications [59]	Use of NHI datasets for AI model training and real-world evidence	Government-maintained NHI database; academic access	Active; multiple AI studies published	Strong domestic dataset utility; reproducibility challenges internationally
U.S.—FDA SaMD AI/ML Action Plan (2023) [60]	Lifecycle regulation of AI/ML medical devices; PCCP for adaptive algorithms	Federal regulatory resources; industry compliance	Implemented; ongoing stakeholder consultations	Transparent lifecycle monitoring; global reference for adaptive AI regulation
EU—Artificial Intelligence Act (2021) [18]	Risk-based classification of AI systems; emphasis on safety and transparency	EU-wide legislative framework; member state enforcement	Adopted; implementation phase	Global benchmark for AI governance; healthcare AI classified as high-risk
WHO—Global Benchmarking Tool (GBT) [61]	Strengthening national regulatory systems for medical products	WHO technical assistance; country participation	Active; applied in >70 countries	Provides comparative regulatory maturity scores; situates Taiwan’s TFDA globally

**Table 4 healthcare-14-00023-t004:** The university-led achievements are driving Taiwan’s smart healthcare industry.

University	Achievements
National Taiwan University (NTU) and NTU Hospital (NTUH)	Co-developed and deployed an AI colonoscopy system for polyp and tumor detection in collaboration with aetherAI and industry partners, achieving 96% clinical accuracy [64]. Documented active participation in national smart healthcare research networks, including disease-specific clinical trial cooperation [65]. NTUH’s pancreatic cancer team developed an AI-assisted computed tomography (CT) interpretation model capable of detecting tumors smaller than 2 cm, which are often missed by conventional diagnostics [66].
National Yang Ming Chiao Tung University (NYCU)	Secured quality management system (QMS) certification for three intelligent medical systems, AI brain-tumor detection, critical-care early-warning platform, and anesthetic transmission puncture needle. This places NYCU among the few Taiwanese institutions holding multiple QMS/device licenses [67]. Introduced the DeepBT Detector for brain tumor detection, the Yu Qu Tong Gong Healthcare Platform for elderly care, and a Healthcare GPT for medical record summarization [68]. Developed the world’s first brain-reading AI-driven walking rehabilitation robot (HopeStride). This intelligent rehabilitation robot, capable of “reading the brain”, now allows stroke survivors and patients with degenerative conditions to execute “brain-driven walking.” It enables active participation, accelerates recovery, reduces therapist workload, and enhances clinical efficiency for rehabilitation physicians [69].
National Cheng Kung University (NCKU) and NCKU Hospital	Showcased 12 AI/precision-medicine teams at the 2024 Taiwan Healthcare Expo, demonstrating end-to-end integration from prevention to treatment [39]. The “Smart Healthcare Paradigm” program and seminars share hospital best practices and outcomes to speed diffusion [32].
Taipei Medical University (TMU) and TMU Hospital (TMUH)	TMU opened Taiwan’s first AI-integrated smart long-term care facility, extending university R&D into real-world eldercare operations [70]. System-level digital innovation across TMU Healthcare System, including smart wards, telemedicine, digital pathology, and precision diagnostics [71].
China Medical University (CMU) and CMU Hospital (CMUH)	Achieved HIMSS AMAM Stage 7 for analytics maturity, underscoring leadership in data-driven care [37]. CMUH reported the rare combination of EMRAM 7, AMAM 7, Design-Implement-Assess-Modify (DIAM) 6, and topped the HIMSS Digital Health Indicator [72]. Developed an AI-powered “intelligent antimicrobial system (iAMS)” that rapidly identifies drug-resistant bacteria, cutting diagnostic time from 72 h to one hour [73].
Kaohsiung Medical University (KMU) and KMU Hospital (KMUH)	Achieved HIMSS EMRAM Stage 7 following Stage 6 in 2023, alongside national smart-hospital quality recognition, and providing the evidence of sustained digital transformation [74]. KMU’s AI Biomedical Research Institute convenes cross-university and industry forums on smart healthcare and long-term care, with applications of generative AI [75].
Chang Gung University (CGU) and Chang Gung Memorial Hospital (CGMH, Linkou)	Unveiled real-time AI ultrasound for infant hip dysplasia, showing university–health-system innovation translating to pediatric screening [76]. The AI Research Center of CGU runs joint smart-medical projects with Chang Gung Medical Foundation and Formosa Plastics Group companies [77].

**Table 5 healthcare-14-00023-t005:** The ICT companies that have invested in Taiwan’s smart healthcare industry.

ICT Company/ Institution	Annual Revenue (USD, Last Year)	Role in Smart Healthcare Industry
Quanta	23.67 billion (2024)	Developed QOCA telemedicine platform, which plays a critical role in enabling remote healthcare services in smart hospitals [82].
ASUS	17.52 billion (2024)	Offers AICS smart healthcare solutions, integrating AI and Internet of Things (IoT) to support clinical environments [83].
Acer Medical (a branch of Acer)	7.77 billion (2023)	Created AI diagnostic tools such as VeriSee DR (for diabetic retinopathy) and VeriSee AMD, along with the TeleMed telemedicine app, enabling remote patient consultation and monitoring [84,85].
BenQ Medical Technology (via Qisda)	6.51 billion (2024)	Advanced the development of medical hardware essential to smart hospital systems [86].
Leadtek	4.29 billion (2024)	Expanded into healthcare products, including Alvital-brand oximeters, wearable sleep monitoring services, and home health devices.
Wistron Medical Technology (via Wistron)	33.83 billion (2024)	Produced the BestShape VS system, a non-contact vital signs sensor that continuously detects respiration, heart rate, and bed presence in elderly patients, providing 24 h real-time monitoring and data display [87,88].
Compal Electronics	29.36 billion (2024)	Introduced the Compal iCare System and BoostFix. The iCare platform supports over 3000 long-term and residential care institutions across Taiwan, streamlining administrative tasks such as uploading data to the NHI Bureau [89,90].
Foxconn (Hon Hai)	212.92 billion (2024)	Leading EMS firms (e.g., Foxconn, Quanta, Inventec, etc.) are deepening investments in smart healthcare, recognizing strong synergies between ICT advancements and healthcare innovation [91].
Inventec	21.60 billion (2024)	Actively engaged in AI-driven smart healthcare partnerships [92,93].
StarFab of ITRI	Not available	While not a commercial ICT company, ITRI plays a pivotal role in fostering smart medical innovation and startups through its StarFab accelerator, providing cross-industry R&D, infrastructure, and incubation resources for ICT-healthcare integration [94,95,96].

**Table 6 healthcare-14-00023-t006:** AI-related medical products developed in Taiwan.

Catalog	Developer (Approval)	Product	Key Features
AI- assisted Medical Imaging Diagnostics	Acer Medical (TFDA)	VeriSee DR	AI-powered diabetic retinopathy screening tool approved by Taiwan’s TFDA [32].
		VeriSee AMD	A TeleMed telemedicine app, enabling remote patient consultation and monitoring (https://www.acer-medical.com/solutions/verisee-amd/ (accessed on 12 November 2025) [84].
		VeriSee Chest	AI-based chest radiograph analysis platform.
	AmCad BioMed (TFDA; FDA)	AmCAD-UT^®^	AI ultrasound analysis for thyroid cancer risk stratification [102,103].
		AmCAD-UO^®^	AI-based sleep apnea diagnostic platform.
		AmCAD-CA	Digital cytopathology analysis system [104].
	Crystalvue (TFDA; FDA; CE)	VISION 700	Ophthalmic diagnostic imaging devices (such as fundus cameras) (https://www.crystalvue.com.tw/en/product.php?act=view&id=35 (accessed on 12 November 2025).
	EverFortune AI (TFDA; FDA)	ChestSuite XR Assessment System	AI Performance in Lung Cancer Detection on CT Thorax.
	aetherAI (TFDA)	aetherAI Hema	Uses digital morphology analysis software for bone marrow smear evaluation. An AI colonoscopy with clinical performance reported at 96% accuracy (https://www.aetherai.com/zh/our-service (accessed on 12 November 2025) [105].
	NCKU Megapro biomedical	iMbody	A sarcopenia screening test predicts mortality among hospitalized cancer patients [106].
	V5med (TFDA; FDA)	V5-MED-LU01	An AI-assisted lung cancer diagnosis module was developed by V5med Inc. and AstraZeneca Taiwan. Starting with low-dose CT image interpretation, the module supports early lesion detection of solitary pulmonary nodules [107].
Data Integration and Analytics Systems	iXensor (TFDA; FDA)	PixoTest^®^	Smartphone-integrated blood glucose monitoring platform (http://www.primemedix.com.ph/products/pixotest-blood-glucose-monitoring-system/ (accessed on 12 November 2025).
		PixoHealth Pass	Digital Health Passport System.
		PixoTest POCT System	Multi-modal point-of-care testing system (including blood glucose, COVID-19 rapid test, smartphone-enabled diagnostics)
	ITRI	MedBobi 2.0	Voice-to-text system for clinical documentation improves efficiency by 10× (https://event.itri.org/CES2025/tech_detail/7/ (accessed on 12 November 2025).
	Wistron Medical Technology (TFDA; FDA)	BestShape VS system	A non-contact vital signs sensor to detect the elderly’s respiration, heart rate, and not-in-bed status without touching them [87].
AI-Powered Virtual Assistants and Intake Avatars	IntoWell Biomedical Technology (collaboration with Meiyo Medical Technology, Samoa) (TFDA)	Dr. AI	Multilingual virtual assistant that captures symptoms and generates SOAP notes (https://www.draiai.com/ (accessed on 12 November 2025).
		SOAP QuickNote	Auto-generates clinical notes, reduces physician documentation workload (https://www.draiai.com/tw/soap-quicknote (accessed on 12 November 2025).
		MedPA AI Avatar	Pre-consultation AI agent that prepares clinical documentation before patient entry.
Smart ward and surgical navigation systems	ITRI (collaborated with hospital partners and various Taiwanese medtech firms)	Smart Bed Sensor System	Monitors pressure distribution, prevents falls and bedsores.
		Surgical Navigation System	Combines AI and imaging for precision and safety in surgery.
		PECOLA robot	Assists elderly adults living alone by combining ambient intelligence with computer vision [108].
	HIWIN Technologies (collaboration with CMUH)	Hiwin Smart Surgical System	Combines robotic arms with AI imaging for precision surgery.
Telemedicine and Cloud-based Systems	NHIA (collaborates with Google)	Google Cloud Diabetes Program	AI analytics for managing 2 million diabetic patients (AI-Powered Healthcare in Taiwan) [36].
	Health2Sync (TFDA)	App and cloud analytics platforms	Diabetes management platform integrated with blood glucose data in collaboration with the NHIA [50].
	Chang Gung Memorial Hospital	Biobank-EMR Integration	AI models for cancer recurrence prediction. [109]
Fubon Financial Administrative automation and insurance intelligence	NHIA	AI Claims Review System	Smart algorithms for billing review and fraud detection (https://www.fubon.com/financialholdings/citizenship/downloadlist/downloadlist_report/Fubon_ESGreport_2021_EN.pdf (accessed on 12 November 2025).

## Data Availability

No new data were created or analyzed in this study.

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
