# Peer review of "Healthcare2026, 14(1), 23;https://doi.org/10.3390/healthcare14010023"

_healthcare, 2025, doi:10.3390/healthcare14010023_

Round 1

Reviewer 1 Report

Comments and Suggestions for Authors

The manuscript presents a timely and relevant overview of Taiwan’s rapidly advancing smart healthcare ecosystem, supported by strong national policy, ICT infrastructure, and medical research capacity. The content is informative, but several refinements can improve clarity, depth, and policy-technology alignment.

  1. Clarify the strategic rationale for prioritizing smart healthcare—such as aging demographics, healthcare workforce shortages, or chronic disease burden.
  2. Add a short explanation of how NHI’s integrated data architecture (e.g., claims data, MediCloud, My Health Bank) uniquely supports AI model development.
  3. Include examples of real-world AI-enabled applications (e.g., predictive analytics, imaging AI, telemedicine) to demonstrate tangible impact beyond policy descriptions.
  4. Strengthen transitions between government policy, academic R&D, and ICT industry collaboration to enhance narrative flow.
  5. Provide brief context for technical terms (e.g., MediCloud functions, NHI data-sharing mechanisms) to improve readability for global audiences.
  6. Discuss remaining barriers—data governance, interoperability standards, regulatory sandboxes, and AI validation guidelines—to give a balanced perspective.
  7. Highlight opportunities in cross-border AI healthcare collaborations where Taiwan’s strong ICT sector may offer global competitiveness.
  8. Suggest emphasizing patient-centric digital transformation, including privacy protection, cybersecurity, and equitable access to smart healthcare technologies.
  9. The following studies are suggested to evaluate and add to the literature review of the manuscript: https://doi.org/10.1109/JIOT.2025.3525623, https://doi.org/10.3389/fbioe.2020.00547, https://doi.org/10.1371/journal.pone.0331464

Reviewer 2 Report

Comments and Suggestions for Authors

Dear Authors,
Thank you for submitting your manuscript to Healthcare. The topic is timely and the manuscript compiles substantial amount of information on Taiwan’s smart healthcare ecosystem. The policy timeline, program summaries, and product tables are particularly informative. However, several important issues need to be addressed before manuscript can be considered for publication.

1. Reference [1] cites 2025 data while text discusses 2024; clarify year and data source.
2. Several TFDA-related citations do not match claims; verify and correct all regulatory references.
3. Manuscript lists programs and achievements without evaluation frameworks or global benchmarking.
4. Tone is overly positive; add discussion of AI failures, regulatory challenges, and reproducibility issues.
5. Table 4 lacks revenue, investment, and export analysis for ICT companies.
6. Explain why Taiwan’s AI-medical products have not achieved strong global competitiveness.
7. State-of-the-art review excludes digital-twin healthcare, essential modern smart-healthcare component.

Reviewer 3 Report

Comments and Suggestions for Authors

This manuscript provides a broad, timely overview of Taiwan’s smart healthcare ecosystem, spanning policy, national programs, academic achievements, ICT industry participation, and concrete AI-powered products. The integration of NHI data assets, ICT strengths, and government policy into a coherent narrative is valuable and will be of interest to readers seeking a national-level case study of AI in healthcare.

Below are suggestions to strengthen the paper:

1. Clarify scope and add a methods/approach section

  • At present the paper reads as a well-curated narrative review / landscape report, but the methodology for selecting policies, programs, institutions, and products is not described.

  • Consider adding a short “Methods” or “Approach” section after the Introduction specifying:

    • How sources were identified (databases, government portals, grey literature, news, conference materials, etc.).

    • Time frame covered.

    • Any inclusion/exclusion criteria (e.g., only TFDA-approved AI devices, only national-level programs, etc.).

  • This will help readers understand how comprehensive and reproducible the overview is and improve perceived scientific rigor.

2. Improve structure and reduce repetition

Overall organization (timeline → national programs → university role → ICT companies → products → conclusions) is logical, but some tightening would help:

  • Sections 2 and 3 both mix high-level narrative with detailed program descriptions. Consider:

    • Keeping Section 2 focused strictly on the historical/policy timeline.

    • Using Section 3 to synthesize and compare major national programs (goals, funding, status, outcomes) rather than repeating similar descriptive phrases.

  • The conclusion repeats several earlier claims (e.g., NUMBEO ranking, leadership position, “AI Island” vision). This could be more concise, emphasizing:

    • Key achievements,

    • Remaining gaps (governance, interoperability, commercialization),

    • Clear forward-looking recommendations.

3. Deepen critical analysis vs. purely descriptive content

The paper is very strong descriptively but lighter on critical appraisal. To increase impact:

  • For major national programs (TPMI, NSTC Smart Healthcare Phase I/II, TSHA, DHP, Healthy Taiwan Deepening Project), briefly discuss:

    • Measurable outcomes to date (e.g., number of deployed systems, patient outcomes, cost savings, regulatory approvals).

    • Key challenges encountered (e.g., cross-institution data sharing, workforce adoption, funding continuity).

  • For AI products (Table 5):

    • Where possible, highlight supporting clinical evidence: study designs, sample sizes, key performance metrics, and whether there are peer-reviewed validation studies versus only regulatory approvals or marketing material.

    • Comment on real-world deployment (pilot vs routine use, single center vs multi-center).

  • Consider adding a short subsection explicitly addressing limitations of Taiwan’s model (e.g., reliance on a single-payer NHI, cultural/regulatory context) and how transferable the lessons are to other countries.

4. Strengthen discussion of data governance, ethics, and workforce readiness

You briefly mention PDPA and “responsible AI” but these are not developed:

  • Add more detail on:

    • Data governance frameworks, consent models, and approaches to secondary use of NHI data for AI.

    • How bias, fairness, and transparency are handled in AI model development and deployment.

    • Workforce implications: training needs for clinicians, nurses, IT staff; how hospitals are addressing AI literacy and role changes.

  • A short table or paragraph comparing Taiwan’s regulatory approach with major reference jurisdictions (e.g., EU AI Act trends, FDA/EMA thinking) would add context.

5. Clarify numbers, consistency, and value chain framing

  • Some numbers could be more clearly attributed and harmonized:

    • Projected AI healthcare market values (e.g., USD 476.5 billion, CAGR 23%) should consistently specify whether they are global or Taiwan-specific; currently this is somewhat ambiguous.

    • NUMBEO ranking is described as “seven consecutive years” and later “eight consecutive years”; ensure internal consistency.

  • The title emphasizes “value chain” from R&D to industry deployment. Consider:

    • Adding a dedicated figure that explicitly maps the value chain (e.g., data → R&D → validation → regulation → commercialization → clinical integration → outcomes).

    • Using that framework consistently to structure subsections (e.g., “Data Foundations”, “R&D and Validation”, “Industrialization and Export”, etc.).

6. Tables and figures: minor refinements

  • Figure 1 and Table 1: The timeline is very useful, but a lot of text is packed into the table. You might:

    • Standardize tense and phrasing.

    • Group items visually (e.g., data infrastructure, regulation, hospital milestones).

  • Tables 3–5: These are a strength of the paper. Consider:

    • Adding a column indicating type of evidence (e.g., “TFDA-approved device; peer-reviewed clinical trial”, “Pilot deployment”, “Commercial, no peer-reviewed data reported”).

    • Checking that all companies and universities in the text appear in the tables (and vice versa) to avoid confusion.

7. Language and style

The manuscript is generally readable but would benefit from careful language editing:

  • Correct minor grammatical issues, awkward phrasing, and typographical artifacts (e.g., “ㄍ industrialization of medical devices”, inconsistent hyphenation such as “smart-healthcare” vs “smart healthcare”).

  • Standardize abbreviations and ensure all are defined at first use (e.g., NHI, NHIA, FHIR, AMAM, EMRAM, DIAM, PDPA).

  • Ensure consistent use of tenses (present vs past) when describing established programs versus future plans.

8. Limitations section

Consider adding an explicit “Limitations” subsection, for example:

  • Potential incompleteness of the landscape overview (rapidly evolving field, focus on national-level programs).

  • Dependence on publicly available documents and grey literature for some initiatives.

  • Limited quantitative outcome data for some AI deployments.

This will help frame the work appropriately as a narrative/landscape review rather than a systematic review.

Overall, this is a well-timed and potentially impactful overview of Taiwan’s smart healthcare ecosystem. With clearer methodological framing, slightly deeper critical analysis, and some language/consistency polishing, it could make a strong contribution to the literature on national AI-in-health strategies.

Round 2

Reviewer 2 Report

Comments and Suggestions for Authors

Well done!